# Prognostic Imaging Biomarkers in Diabetic Macular Edema Eyes Treated with Intravitreal Dexamethasone Implant

**DOI:** 10.3390/jcm12041303

**Published:** 2023-02-06

**Authors:** Eliana Costanzo, Daniela Giannini, Daniele De Geronimo, Serena Fragiotta, Monica Varano, Mariacristina Parravano

**Affiliations:** 1IRCCS—Fondazione Bietti, Rome, Italy; 2Ophthalmology Unit, Department NESMOS, Sant’ Andrea Hospital, University of Rome “La Sapienza”, Rome, Italy

**Keywords:** biomarkers, OCT, OCTA, predictive models, dexamethasone implant, DME

## Abstract

Background: The aim was to evaluate predictive value of baseline optical coherence tomography (OCT) and OCT angiography (OCTA) parameters in diabetic macular edema (DME) treated with dexamethasone implant (DEXi). Methods: OCT and OCTA parameters were collected: central macular thickness (CMT), vitreomacular abnormalities (VMIAs), intraretinal and subretinal fluid (mixed DME pattern), hyper-reflective foci (HRF), microaneurysms (MAs) reflectivity, ellipsoid zone disruption, suspended scattering particles in motion (SSPiM), perfusion density (PD), vessel length density, and foveal avascular zone. Responders’ (RES) and non-responders’ (n-RES) eyes were classified considering morphological (CMT reduction ≥ 10%) and functional (BCVA change ≥ 5 ETDRS letters) changes after DEXi. Binary logistic regression OCT, OCTA, and OCT/OCTA-based models were developed. Results: Thirty-four DME eyes were enrolled (18 treatment-naïve). OCT-based model combining DME mixed pattern + MAs + HRF and OCTA-based model combining SSPiM and PD showed the best performance to correctly classify the morphological RES eyes. In the treatment-naïve eyes, VMIAs were included with a perfect fit for n-RES eyes. Conclusion: The presence of DME mixed pattern, a high number of parafoveal HRF, hyper-reflective MAs, SSPiM in the outer nuclear layers, and high PD represent baseline predictive biomarkers for DEXi treatment responsiveness. The application of these models to treatment-naïve patients allowed a good identification of n-RES eyes.

## 1. Introduction

Diabetic macular edema (DME) is an important complication of diabetic retinopathy, representing the main cause of visual impairment in diabetic patients [1].

The current/modern approach to center-involved DME consists of the intravitreal injection of anti-vascular endothelial growth factor (VEGF) agents and/or steroid drugs, in particular, dexamethasone implant (DEXi) [2].

The clinician’s choice between these different drugs is usually based on several factors that consider the patient’s compliance and motivation, age, and some general and ophthalmological comorbidities [2]. Furthermore, several imaging biomarkers have been identified as particularly useful to guide the therapeutical option choice [3], but a univocal consensus has not yet been reached on this topic.

Optical coherence tomography (OCT) and OCT angiography (OCTA) allow non-invasive recognition of imaging features that define a “DME profile”, distinguishing an inflammatory DME pattern characterized by the presence of subretinal fluid (SRF) and a high number of hyper-reflective foci (HRF) [4,5]. Furthermore, numerous imaging (OCT and OCTA) biomarkers have been identified in DME eyes as predictive of treatment response or disease progression, such as the retinal thickening characteristics [3,6,7], the presence, position, and internal reflectivity of microaneurysms (MAS) [8,9], the vitreomacular interface abnormalities (VMIAs) [6], the ellipsoid zone/external limiting membrane (EZ/ELM) integrity/disruption [7], the presence of nonperfusion areas (NPAs) [10,11], the foveal avascular zone (FAZ) abnormalities in terms of circularity and dimension [12], the involvement of deep capillary complex (DCP) [10,13], and the suspended scattering particles in motion (SSPiM) [14,15].

This study aimed to explore the influence of baseline OCT and OCTA parameters, isolated or combined, on the DEXi response in diabetic macular edema eyes, integrating the information provided by these two different techniques through statistical models.

## 2. Materials and Methods

### 2.1. Study Participants

In this study, consecutive type 2 diabetic patients affected by DME treated with DEXi were retrospectively collected and analyzed at the Department of Ophthalmology, IRCCS-Fondazione Bietti, Rome.

This observational study was approved by the Institutional Review Board of the IRCCS-Fondazione Bietti and followed the tenets of the Declaration of Helsinki. Written informed consent was obtained from all participants.

Each patient received a single DEXi (0.7 mg, Ozurdex; Allergan, Inc., Irvine, CA, USA) to treat DME, as per clinical practice. Inclusion criteria were age ≥ 18 years, type 2 diabetes mellitus, naïve or previously treated DME eyes (at least 3 months after the last anti-VEGF injections or 6 months after the last DEXi or other steroid drug injection), and a minimum follow-up (FU) period of 4 months after treatment, with full imaging protocol. Exclusion criteria were macular edema secondary to other causes (e.g., retinal vein occlusion), significant lens opacity, graded above NO3 or NC3 [16], refractive error equal to or lower than ± 2 diopters, measured as spherical equivalent, absence of moderate to dense corneal opacities, refractive surgery, glaucoma or ocular hypertension, medical history of intraocular inflammation, poor-quality images with a signal strength index (SSI) lower than 7 for the PlexElite OCTA, or the presence of significant motion artifacts (seen as large dark or grey lines on the enface angiograms) that impaired the slabs’ quality.

Patients received a complete ophthalmological examination, which included the measurement of best corrected visual acuity (BCVA) using Early Treatment of Diabetic Retinopathy Study (ETDRS) visual charts, intraocular pressure (IOP), and dilated fundus examination. All patients were imaged by spectral domain (SD) OCT using Spectralis (Heidelberg Engineering, Heidelberg, Germany) and by swept source (SS) OCTA using PlexElite 9000 (Carl Zeiss Meditec Inc., Dublin, CA, USA) device.

Patients’ charts, BCVA, SD-OCT, and SS-OCTA parameters (see below) at baseline were reviewed, and their influence on treatment response 4 months after DEXi was analyzed.

### 2.2. Imaging Protocol

SD-OCT images were acquired at baseline, monthly, and at the end of follow-up (4 months after DEXi). The scans were obtained using Spectralis (Heidelberg Spectralis version 1.10.2.0, Heidelberg Engineering, Heidelberg, Germany) with a raster scan using an acquisition protocol of a minimum 20 × 15 degree pattern constituting 19 consecutive B-scans and a macular map centered on the fovea.

All raster B-scan images were checked for errors in automatic segmentation, and a manual correction was made in case of segmentation errors.

SS-OCTA images were acquired using the PlexElite 9000 device (software version 1.7.027959; Carl Zeiss Meditec, Inc., Dublin, CA, USA), which uses a swept laser source with a central wavelength of 1050 nm and a bandwidth of 100 nm [17]. This instrument has an axial resolution of approximately 5 μm and a lateral resolution estimated at approximately 14 μm. OCTA images were acquired using a 6 × 6 mm volume captured with FastTrac eye motion correction software. The built-in segmentation software automatically segmented the whole retina slab, the superficial vascular complex (SVC) slab, and the deep capillary plexus (DCP); the whole retina vasculature slab includes automatic segmentation from the inner limiting membrane (ILM) up to 70 mm above the retinal pigment epithelium (RPE) [18]; the SVC was segmented between the ILM and the inner plexiform layer (IPL); for the DCP, the upper limit was the IPL and the lower was defined by the outer plexiform layer (OPL). The correctness of retinal boundaries was checked by two single experienced examiners (EC and MP); in case of misplacing, the segmentation was manually adjusted.

### 2.3. SD-OCT Parameters

The quantitative OCT parameters were the central macular thickness (CMT), automatically measured using instrument software, and the number of hyper-reflective foci (HRF).

The number of HRF was manually counted in the parafoveal region, using the ETDRS grid integrated into the Spectralis, identifying the correct area.

The qualitative parameters were the pattern of DME, the presence and the internal reflectivity of the microaneurysms (MAs) (i.e., hyporeflective, hyperreflective, or mixed), the presence of vitreomacular interface abnormalities (VMIAs), and the disruption of ellipsoid zone (EZ). The DME pattern was defined as “mixed” if intraretinal cysts (IRC) were detected in combination with subretinal fluid (SRF). The MAs and the VMIAs were evaluated by scrolling all raster B-scans. The EZ disruption was evaluated in a linear scan passing through the fovea.

### 2.4. OCTA Parameters and Analysis

The quantitative OCTA parameters evaluated at baseline were the foveal avascular zone (FAZ) area, the perfusion density (PD), and the vessel length density (VLD) at the whole retina vasculature slab.

The qualitative OCTA parameters included the presence of suspended scattering particles in motion (SSPiM), FAZ erosion, and nonperfusion areas (NPAs) in the whole retina vasculature slab, as well as the MAs visualization both in the SVC and DCP.

The quantitative OCTA parameters were measured as follows: the OCTA whole retina vasculature slab was opened on FIJI (an expanded version of ImageJ: 2.0.0-rc-69/1.52p; National Institutes of Health) [19], the FAZ border was manually outlined, and the surface area, expressed in mm^2^, was measured as previously reported [20]. In addition, the slabs were binarized to generate a black and white image for measuring the PD, and then the images were skeletonized to calculate VLD [20,21]. The PD defines the ratio of the area occupied by the vessels divided by the total area, providing complete vasculature information in terms of size and length [21]; the VLD defines the total vessel length divided by the total number of pixels in the analyzed skeletonized image [22] and may be more sensitive to the microvasculature changes [21].

The OCT and OCTA qualitative parameters were independently evaluated by two expert readers (EC and MP), and the interclass correlation coefficient (ICC) was calculated. In case of disagreement, a third reader (DDG) assigned the final grade.

All OCT and OCTA parameters included in our analysis were evaluated at baseline to explore their influence on the DEXi 4-months response.

### 2.5. Criteria for Groups’ Classification

The study population was classified either morphologically or morpho-functionally.

The morphological response was staged according to the Protocol T and Protocol I definitions: “improvement” if CMT decreased at least 10% and “no improvement” for a CMT variation <10% since baseline visit [5,23]. Using the CMT value, the change in macular thickness (Δ CMT) between baseline and 4 months after treatment was calculated. Based on this definition, we subdivided our sample into two groups: responder (RES) if a CMT reduction of at least 10% was recorded and non-responder (n-RES) in all other cases.

Additionally, as a secondary outcome of our study, we evaluated the changes in BCVA after treatment ≥5 ETDRS letters [24] in order to perform a morpho-functional classification considering CMT + BCVA changes from baseline.

A sub-analysis of treatment-naïve eyes was also conducted.

### 2.6. Statistics

Statistical evaluation was performed using SPSS (IBM Corp. Released 2017. IBM SPSS Statistics for Windows, version 25.0. Armonk, NY, USA: IBM Corp.). Continuous variables, including age, BCVA ETDRS letters score, and instrument parameters were expressed as mean ± standard deviations (SD), while categorical variables were expressed as frequencies.

The ICC was calculated to estimate the absolute agreement between the two expert readers’ (EC and MP) grading on OCT DME mixed pattern, EZ disruption, VMIAs, MAs internal reflectivity, and OCTA FAZ erosion, presence of NPAs, SSPiM, and MAs visualization (ICC < 0.5, poor reliability; 0.5 < ICC < 0.75, moderate reliability; 0.75 < ICC < 0.9, good reliability; ICC > 0.9, excellent reliability) [25].

The normal data distribution was tested using the one-sample Kolmogorov–Smirnov test. The independent sample 𝑡-test and the Mann–Whitney test were used to compare the parameter values between the two groups. A chi-square test or a Fisher exact test two sides, as appropriate, was performed to investigate the relationship between the groups and the clinical categorical variables.

A binary logistic regression model was applied using the OCT and OCTA variables as independent explanatory variables (i.e., predictors) to classify responder (RES) and non-responder (n-RES) eyes. A threshold of 0.5 was chosen. The variance inflation factor (VIF), which assesses how much the variance of an estimated regression coefficient increases if the predictors are correlated, was used to assess multicollinearity; only variables with a VIF > 1 and VIF < 10 were considered as covariates.

To evaluate and select the final developed models, we measured the complexity by the Akaike Information Criterion (AIC) and the Bayesian Information Criteria (BIC), while the accuracy of estimated probability was measured by the Brier’s score [26,27]. Lower values of AIC, BIC, and Brier’s score indicate better goodness of fit, while higher area under the curve (AUC) values indicate better discriminative ability. The model performance to distinguish between RES and n-RES eyes was measured by the receiver operating characteristic (ROC) analysis with the AUC.

Statistically significant differences were set at *p*-value < 0.05 for all the tests performed.

## 3. Results

A total of 34 eyes (18 treatment-naïve) of 30 DME patients (10 females and 20 males) were enrolled; for four DME patients (one female and three males), we included both eyes.

The mean ± SD patient age was 67.4 ± 9.3 years (range 46–81 years). BCVA at baseline was 61.3 ± 14.5 ETDRS letters (20/50 Snellen equivalent, ranging from 20/20 to 20/400). The mean ± SD at baseline CMT was 544.3 ± 137 µm, as determined on the macular map. All included eyes showed moderate non-proliferative diabetic retinopathy and no cases of proliferative diabetic retinopathy had been enrolled.

By following the morphological response classification into two subgroups (RES and n-RES), 20 out of 34 eyes were classified as the RES group (58.8%) and 14 out of 34 eyes were classified as the n-RES group (41.2%).

The RES eyes group (20 eyes) included 18 DME patients (7 females and 11 males) and the n-RES eyes group (14 eyes) included 14 DME patients (3 females and 11 males). Of note, for two DME patients (one female and one male), we included both eyes in the RES group; for the other two DME patients (both males), one eye was RES and the fellow eye n-RES to dexamethasone implant.

Demographic, OCT, and OCTA characteristics of RES vs. n-RES eyes at baseline are detailed in Table 1. No statistically significant differences were found between groups at baseline (all *p*-values > 0.05).

As secondary outcomes, the morpho-functional (MF) response classification identified 24 out of 34 eyes as MF-RES eyes and 10 out of 34 eyes as MF-nRES eyes.

An ICC agreement of 0.985 was found between two readers (EC and MP) on the parameter qualitative evaluations.

### Model Results

We used a binary logistic regression model including OCT and OCTA (isolated and combined) baseline parameters as predictive variables for a 4-month DEXi response.

Different combinations of OCT, OCTA, and OCT/OCTA parameters were tested, exploring all variables collected for each patient. AIC, BIC, and Brier’s scores were calculated to choose the best statistical models.

Nine statistical models were derived, distinguishing between models for morphological classification (models 1, 2, and 3) and morpho-functional classification (models 4, 5, and 6) of the overall population, while three additional models were derived for treatment-naïve eyes (models 7, 8, and 9). The OCT-based models for the overall population (morphological and morpho-functional classification) (models 1 and 4) included DME mixed pattern + MAs internal reflectivity + number of parafoveal HRF. The OCT-based model for the treatment-naïve eyes (model 7) also included the VMIAs as an additional variable. The OCTA-based models for all eyes (morphological overall and treatment-naïve and morpho-functional classification) (models 2, 5, and 8) included PD of the whole retina vasculature slab and SSPiM. The OCT/OCTA-based models combined the OCT and OCTA variables considered in the isolated models for all groups. The variables included in each model and the values of AIC, BIC, and Brier’s score are reported in Table 2.

Table 3 reports the models’ performance analysis, where the sensitivity was related to RES detection and the specificity to the n-RES.

Model 1 correctly classified 80% of RES and 64.29% of n-RES eyes and the overall accuracy was 72.1% (*p* = 0.006). In model 2, the overall AUC was 55.7% and the performance of this model was not statistically significant, *p* = 0.390. Model 3 correctly classified 70% of RES and 64.3% of n-RES eyes, with an overall accuracy of 67.1% (*p* = 0.043).

The performance for models 4, 5, and 6 (based on the sample of morpho-functional classification) was not statistically significant.

Models from 7 to 9, considering the treatment-naïve eyes, were all statistically significant (*p* < 0.001 for models 7 and 9, *p* = 0.005 for model 8), with a perfect fit detection of RES and n-RES eyes by the combination of OCT and OCTA parameters.

By the analysis of the combination of these models, the RES eyes were more likely to show a DME mixed pattern (IRC + SRF) and hyper-reflective MAs associated with a higher number of HRF in the parafoveal region at baseline compared with n-RES correctly classified by the model. Furthermore, RES eyes showed SSPiM in the ONL (72.2% of eyes), DCP (44.4%), and SVC (22.2%) but, also, a higher PD compared with n-RES (44.34 ± 1.29 vs. 40.39 ± 0.9, respectively).

The combination of OCT and OCTA parameters (model 3) showed a good performance but lower than model 1 alone.

The subgroup analysis of treatment-naïve eyes allowed a good identification of n-RES eyes through OCT- and OCTA-based models, reaching a perfect fit of RES and n-RES detection in the combined model (model 9). The n-RES eyes at baseline were characterized by the presence of VMIAs, mixed/hyporeflective MAs, lower number of parafoveal HRF, and absence of SRF at baseline compared with RES eyes correctly classified by model 7. Regarding OCTA parameters (model 8), n-RES eyes likely presented SSPiM in the ONL (50% of eyes) and in the SVC and DCP (16.67% of eyes for both) and showed a lower PD compared to RES eyes (43 ± 2.37 vs. 45 ± 1.04).

## 4. Discussion

The present study explored the baseline OCT and OCTA variables that may predict the DEXi response in DME eyes. All the parameters analyzed are known to be differently associated with treatment response, but we aimed to understand whether some variables could be more predictive than others by using binary logistic regression models.

The OCT-based model took into account different parameters that can be usually found in DME eyes, such as the presence of a mixed fluid pattern (IRC + SRF), parafoveal HRF, the internal reflectivity of MAs, the VMIAs, and the EZ disruption, discovering that not all are equally important within predictive models.

We found that DME eyes responsive to DEXi showed most likely the presence of both intraretinal and subretinal fluid on OCT B-scans at baseline. This finding is in line with several previous studies in which the foveal neuroretinal detachment was defined as an inflammatory biomarker predictive of a better response to steroid agents [4,5,28]. Vujosevic et al. [4] defined an inflammatory pattern of DME characterized by the presence of SRF, a high number of HRF, and areas of increased fundus autofluorescence. Among these factors, eyes with SRF at baseline exhibited a greater decrease in CMT (morphological response) compared with those without SRF at baseline. This pattern seemed to be most responsive to DEXi compared to anti-VEGF agents, suggesting a targeted therapy considering these parameters [4]. Another interesting real-world study [7] reported that eyes with DME mixed pattern had a DME recurrence at ≥ 6 months from the first DEXi compared with eyes with only IRC that showed the earliest DME recurrence, confirming the best response to DEXi when SRF exists. Zur et al. [29] also demonstrated the presence of SRF as a predictive biomarker for a better morpho-functional response to DEXi. The authors found a more remarkable BCVA improvement after DEXi in the case of DME with SRF. Moreover [29], the authors also reported that eyes without HRF showed a better DEXi response. Contrariwise, in our study, RES eyes showed a higher number of parafoveal HRF than n-RES eyes correctly classified by the models for both morphological and morpho-functional classification. Our results agree with other studies [4,30,31] in which a better morphological response to intravitreal anti-VEGF and steroids was associated with the presence of HRF. In addition, a greater HRF reduction after DEXi compared with anti-VEGF treatment has been documented [28,32,33], confirming the HRF as a biomarker of inflammatory DME pattern [4,5,34].

The role of HRF in the prediction of treatment response is still unclear. A recent review [35] analyzing 36 studies (11 prospective and 25 retrospective) with low or moderate risk of bias concluded that it was unclear whether HRF predicts the treatment outcome in patients with DME. In our model, a high number of parafoveal HRF, in association with the presence of SRF and hyper-reflective MAs, represented significant predictors for the DEXi response.

The third significant parameter in our OCT-based models was the internal reflectivity of MAs. Parravano et al. [9] have described differences in the internal reflectivity of MAs, suggesting different patterns of blood flow dynamic distinguishing between hyporeflective and hyper-reflective MAs. The hyper-reflective pattern at baseline was found to be strongly associated with extracellular fluid accumulation in a 1-year follow-up study [8,9], hypothesizing that the hyper-reflective MAs could have a higher inflammatory factor component, responsible for blood–retinal barrier (BRB) changes [8]. Furthermore, the presence of hyper-reflective MAs could lead to a good response to anti-VEGF or steroids due to their high vascular flow within [8]. In our series, RES eyes showed more often hyper-reflective MAs than n-RES in both the overall population and treatment-naïve subgroup, further corroborating their influence on a better treatment response.

The subgroup analysis of treatment-naïve eyes allowed the identification of VMIAs as an additional parameter in the OCT-based model. The abnormalities in the vitreomacular interface have already been reported as a negative predictive factor of treatment response [36,37]. Despite this, our analysis represented a step forward in the characterization of an n-RES phenotype characterized by the absence of SRF, VMIAs, mixed/hyporeflective MAs, and a lower number of parafoveal HRF at baseline. These results were consistent with those obtained from the OCT-based models on the overall population, confirming the importance and the weight of the variables explored and derived from our models.

Our predictive models also investigated several OCTA parameters (SSPiM, PD, VLD, FAZ area, FAZ erosion, and MAs’ visualization in SVC and DCP), of which the most representative were SSPiM and PD. Our results showed that RES eyes were more likely to have SSPiM in the outer retinal layers than in the inner retina (72.2% in ONL, 44.4% in DCP, and 22.2% in SVC) and had a higher PD compared with n-RES correctly classified.

In a recent paper by our group [14], the presence of SSPiM and its pyramidal stratification from outer to inner retinal layers indicates a severe breakdown of BRB. In the present analysis, the SSPiM, considered alone, did not reach the power to predict the DEXi response but, when associated with other OCT parameters and PD on OCTA, it was able to characterize only the RES eyes in statistically significant models. The RES eyes showed the highest prevalence of SSPiM in the outer retina, with a partial sparing of inner retina indicating that, despite the damage of BRB, typical of DME eyes, a good response to steroid agents could be, however, expected. Given the pyramidal stratification of SSPiM, it may be conceivable that the BRB damage is more severe when the inner retinal layers are involved. Therefore, eyes with SSPiM confined within the outer retinal layers can still be more responsive to the treatment. A relatively small percentage of SSPiM within the SVC in RES eyes may further reinforce this hypothesis (22% vs. 78% in the ONL). The result of this combined model agrees with a theory for the SSPiM being closely related to the number of HRF and being considered the product of inner BRB breakdown, and, in fact, the hyper-reflective cystoid spaces often colocalize with HRF [38].

We have also found that the RES eyes showed a higher PD measured in the whole retina vasculature slab compared to n-RES eyes. A previous study reported that a high baseline PD value in the DCP was significantly correlated with a better baseline VA [39]. Another study reported a lower vessel density (VD) in the DCP in the poor-response group, considering both morphological and functional parameters [40]. All these previous data are consistent with the results of our OCTA-based models, suggesting that the retinal microvascular parameters could predict the treatment response in DME and help optimize clinical outcomes [40].

An interesting consideration concerns the weight of the OCT parameters, which seemed to be greater than the OCTA parameters in our models. The performance of combined models, particularly model 3, was lower than the models including only the OCT variables, indicating that the predictive value of the OCT-based model did not significantly improve by adding the OCTA factors. In fact, the OCTA-based models alone were not significant in the overall population but only in the treatment-naïve eyes. This finding could reasonably be explained with potential confounders that may alter the OCTA metrics in previously treated eyes.

Interestingly, the morpho-functional classification of our population did not lead to significant predictive models. We hypothesize that the complexity of the functional evaluation of diabetic eyes cannot be based only on the measurement of the BCVA, as different degrees of DME may result in a high variability of visual acuity [41]. Further studies including systemic parameters or more sophisticated functional techniques should be considered for defining more accurate predictive models.

The main limitation of our study was represented by the small sample size for the explorative nature of our study, focusing on the predictive value of different OCT and OCTA biomarkers at baseline. Another limitation was the lack of consideration of the metabolic parameters within the variables that can influence the treatment response. However, we aimed to analyze real-world data considering objective variables that can guide the clinician in predicting DEXi response. In our opinion, the promotion of glycemic control still represents one of the most important factors influencing the treatment response and disease progression of diabetic patients with ocular involvement.

Further analysis with a higher number of patients is needed to validate our results, as the statistical models explored in our study could be potentially useful for artificial intelligence in the construction of a strong predictive model for DME eyes to treat with DEXi.

In conclusion, this study demonstrated the predictive value of baseline OCT and OCTA parameters in DME eyes treated with DEXi using statistical logistic regression models. By analyzing the models, the presence of DME mixed pattern, a high number of parafoveal HRF, hyper-reflective MAs, SSPiM in the outer nuclear layers, and high PD represent baseline predictive biomarkers for DEXi treatment responsiveness.

## Figures and Tables

**Table 1 jcm-12-01303-t001:** Baseline characteristics of responder (RES) and non-responder (n-RES) eyes.

	RES (Mean ± SD)	n-RES (Mean ± SD)	*t*-Test/Mann Whitney U Test, *p*
Age	70.45 ± 8.84	64.36 ± 9.078	0.059
BCVA (ETDRS letters)	77.25 ± 14.91	74.29 ± 15.42	0.535
CMT	567.20 ± 145.79	511.57 ± 120.95	0.250
4 month follow-up CMT	404.25 ± 130.03	512.79 ± 131.43	0.023 *
OCT qualitative parameters	RES N = 20 (%)	n-RES N = 14 (%)	Chi square/Exact Fisher test, *p*
DME Mixed pattern (IRC + SRF)	8/20 (40%)	5/14 (35.7%)	0.800
EZ disruption	15/20 (75%)	10/14 (71.4%)	0.816
VMIAs	9/20 (45%)	7/14 (50%)	0.774
MAs internal reflectivity			
Hyper	12/20 (60%)	8/14	0.251
Hypo	3/20 (15%)	0/14	0.868
Mixed	5/20 (25%)	6/14	0.273
OCT quantitative parameters	RES N = 20 (%)	n-RES N = 14 (%)	*t*-test/Mann Whitney U test, *p*
HRF (n.)	42.85 ± 30.24	29.50 ± 24.14	0.180
OCTA qualitative parameters	RES N = 20 (%)	n-RES N = 14 (%)	Chi square/Exact Fisher test, *p*
FAZ erosion	9/20 (45%)	7/14 (50%)	0.774
NPAs	15/20 (75%)	8/14 (57.1%)	0.273
SSPiM			
ONL	5/20 (25%)	4/14 (28.6%)	0.382
ONL/DCP	4/20 (20%)	1/14 (7.1%)	0.379
ONL/DCP/SVC	4/20 (20%)	2/14 (14.3%)	1.000
MAs visualization			
DCP	18/20	14/14	0.501
SCP	10/20	8/14	0.681
DCP/SVC	9/20	8/14	0.486
OCTA quantitative parameters (whole retina slab)	RES N = 20 (%)	n-RES N = 14 (%)	*t*-test/Mann Whitney U test, *p*
FAZ area (mm^2^)	0.40 ± 0.21	0.41 ± 0.12	0.854
PD (%)	44.13 ± 1.40	44.00 ± 2.08	0.835
VLD (%)	11.71 ± 0.95	11.83 ± 1.25	0.540

* Significant value. RES: responder eyes (morphological classification), n-RES: non-responder eyes (morphological classification), BCVA: best corrected visual acuity (ETDRS letters), CMT: central macular thickness, OCT: optical coherence tomography, DME: diabetic macular edema, IRC: intraretinal cysts, SRF: subretinal fluid, EZ: ellipsoid zone, VMIAs: vitreomacular interface abnormalities, MAs: microaneurysms, HRF: hyper-reflective foci, OCTA: optical coherence tomography angiography, FAZ: foveal avascular zone, NPAs: nonperfusion areas, SSPiM: suspended scattered particles in motion, ONL: outer nuclear layer, DCP: deep capillary complex, SVC: superficial vascular complex, PD: perfusion density, VLD: vessel length density.

**Table 2 jcm-12-01303-t002:** Model’s comparing analysis.

	AIC	BIC	Brier’s Score
Morphological classification (ΔCMT ≥ 10%)			
Model 1 OCT (DME Mixed pattern + MAs reflectivity + n.HRF parafoveal)	49.546	57.178	0.384
Model 2 OCTA (SSPiM + PD)	54.369	62.001	0.357
Model 3 OCT/OCTA (model 1 + model 2)	55.856	69.593	0.395
Morpho-functional classification (ΔCMT ≥ 10% or ΔETDRS ≥ 5 Letters)			
Model 4 OCT (DME Mixed pattern + MAs reflectivity + n.HRF parafoveal)	47.343	54.975	0.516
Model 5 OCTA (SSPiM + PD)	48.544	56.175	0.515
Model 6 OCT/OCTA (model 4 + model 5)	53.02	66.757	0.531
Treatment-naïve subgroup—Morphological classification (ΔCMT ≥ 10%)			
Model 7 OCT (DME Mixed pattern + VMIAs + MAs reflectivity + n.HRF parafoveal)	20.692	26.034	0.422
Model 8 OCTA (SSPiM + PD)	27.803	32.255	0.331
Model 9 OCT/OCTA (model 7 + model 8)	20.000	28.904	0.500

AIC: Akaike Information Criterion, BIC: Bayesian Information Criteria, ΔCMT: changes in central macular thickness, SD-OCT: spectral domain optical coherence tomography, DME: diabetic macular edema, mixed pattern: intraretinal and subretinal fluid, MAs: microaneurysms, n.HRF: number of hyper-reflective foci, OCTA: optical coherence tomography angiography, ΔETDRS: changes in ETDRS letters, SSPiM: suspended scattering particles in motion, PD: perfusion density, VMIAs: vitreomacular interface abnormalities.

**Table 3 jcm-12-01303-t003:** Models’ performance analysis.

Target Sensitivity >> Responder Specificity >> No Responder	AUC (95% CI)	Sensitivity (95% CI)	Specificity (95% CI)	*p*-Value
Morphological classification (ΔCMT ≥ 10%)				
Model 1 (SD-OCT)	72.1% (52.2–86.1)	80.0% (56.3–94.3)	64.29% (35.1–87.2)	0.006 *
Model 2 (SS-OCTA)	55.7% (37.7–72.7)	90.0% (68.3–98.8)	21.4% (4.7–50.8)	0.390
Model 3 (SD-OCT + SS-OCTA)	67.1% (49.0–82.2)	70.0% (45.7–88.1)	64.3% (35.1–87.2)	0.043 *
Morpho-functional classification (ΔCMT ≥ 10% OR ΔETDRS ≥ 5 Letters)				
Model 4 (SD-OCT)	55.0% (37.1–72.0)	100.0% (85.8–100.0)	10% (0.3–44.5)	0.317
Model 5 (SS-OCTA)	62.9% (44.7–78.8)	95.8% (78.8–99.9)	30% (6.7–65.2)	0.103
Model 6 (SD-OCT + SS-OCTA)	62.9% (44.7–78.8)	95.8% (78.8–99.9)	30% (6.7–65.2)	0.103
Treatment-naïve subgroup (ΔCMT *≥* 10%)				
Model 7 (SD-OCT)	83.3% (58.6–96.4)	88.89% (51.8–99.7)	77.8% (40.0–97.2)	<0.001 *
Model 8 (SS-OCTA)	77.8% (52.4–93.6)	88.89% (51.8–99.7)	66.7% (29.9–92.5)	0.005 *
Model 9 (SD-OCT + SS-OCTA)	100.0% (81.5–100.0)	100.0% (66.4–100.0)	100.0% (66.4–100.0)	<0.001 *

* Significant value. AUC: area under curve, CI: confidence interval, ΔCMT: changes in central macular thickness, SD-OCT: spectral domain optical coherence tomography, SS-OCTA: swept source optical coherence tomography angiography, ΔETDRS: changes in ETDRS letters.

## Data Availability

The data used to support the findings of this study are available from the corresponding author upon request.

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
