# Peer review of "Prognostic Imaging Biomarkers in Diabetic Macular Edema Eyes Treated with Intravitreal Dexamethasone Implant"

_jcm, 2023, doi:10.3390/jcm12041303_

Round 1

Reviewer 1 Report

Was there any difference between responders and non responders in gender distribution, systemic control (BP, glycemic and lipid), DR status and DRIL on OCT?

Inclusion criteria were naïve or previously treated DME eyes.  For treated (non naïve) eyes,

a)      What were the criteria for shifting to dexamethasone from antiVEGF? Were they non responders to anti VEGF?

b)      How many patients were shifted from anti vegf?

c)       What was the mean number of injections and range of duration of treatment  by anti VEGF in those patients?

d)      How did the mean number of injection and duration of treatment by anti vegf influence the response of those eyes to dexamethasone?

Author Response

We would like to thank the reviewer for the suggestions and comments on our manuscript.

Was there any difference between responders and non-responders in gender distribution, systemic control (BP, glycemic and lipid), DR status and DRIL on OCT?

R: We thank the reviewer for this comment. We included 34 eyes of 30 DME patients (10 females and 20 males), for 4 DME patients (1 female and 3 males) we included both eyes. The RES eyes group (20 eyes) included 18 DME patients (7 females and 11 males), the n-RES eyes group (14 eyes) included 14 DME patients (3 females and 11 males). Of note, for 2 DME patients (1 female and 1 male) we included both eyes in the RES group, for the other 2 DME patients (both males) 1 eye was RES and the fellow eye n-RES to dexamethasone implant. We have added this information in the results section.

Unfortunately, we have only partially information regarding systemic and/or glycemic control and for this reason we did not include these parameters in our statistical models. As we wrote in the discussion, we are convinced that the promotion of glycemic control still represents one of the most important factors influencing the treatment response and disease progression of diabetic patients with ocular involvement.

All included eyes showed moderate non-proliferative diabetic retinopathy, no cases of proliferative diabetic retinopathy have been enrolled. We have added this information in the results section.

The evaluation of the status of inner retina represents an important factor that could affect the DME treatment response, as the reviewer highlighted with this comment. All included eyes showed important macular edema with intraretinal cysts that have influenced the organization of inner retinal layers, for this reason we did not include the DRIL in the OCT baseline parameters for the current analysis.

Inclusion criteria were naïve or previously treated DME eyes.  For treated (non-naïve) eyes,

  1. What were the criteria for shifting to dexamethasone from anti-VEGF? Were they non responders to anti-VEGF?
  2. b)      How many patients were shifted from anti-vegf?
  3. c)       What was the mean number of injections and range of duration of treatment by anti VEGF in those patients?

R a-b-c: We would like to thank the reviewer for this interesting comment. In our study we enrolled 18 treatment-naïve eyes and 16 previously treated eyes. 11 out of 16 eyes were previously treated with anti-VEGF and 5 out of 16 were previously treated with dexamethasone implant. Of 11 eyes previously treated with anti-VEGF, 7 were classified as RES eyes and 4 were classified as n-RES eyes in our models.

In our population the decision to switch from anti-VEGF to dexamethasone has been guided to the OCT morphological response (CMT reduction at least of 10%) and to the patients’ motivation, age, systemic condition, status of lens and the presence of ophthalmological comorbidities.

11 patients were previously treated, 4 of which were non-responders to anti-VEGF (3-6 IVT in the last 6 months without a 10% CMT reduction), 1 patient had a cardiovascular accident, 6 patients were switched to reduce the burden of injections, improving patients’ compliance.

In our series, 11 out of 16 eyes were previously treated with anti-VEGF. Of 11 eyes previously treated with anti-VEGF, 7 were classified as RES eyes and 4 were classified as n-RES eyes in our models.

  1. d)      How did the mean number of injection and duration of treatment by anti vegf influence the response of those eyes to dexamethasone?

R d: Thank you for this interesting consideration. The results from our series, showed that a similar proportion of patients previously treated with anti-VEGF were included in both RES and n-RES eyes: 7 out of 20 eyes for RES and 4 out of 14 eyes for n-RES with a high heterogeneity concerning the drug and the treatment protocol. This paucity and heterogeneity of data did not allow to perform a statistical analysis to explore if the mean n. of injection and treatment duration could influence the dexamethasone response.

Reviewer 2 Report

The study of Costanzo et al shows that, by taking into account a number of OCT and OCTA parameters and after setting up specific binary logistic regression to analyze OCT, OCTA and OCT/OCTA-based models, it was possible to identify a pool of baseline biomarkers of use for the prediction of responsiveness to dexamethasone implant treatment.

Comments to the manuscript

This a well designed, well performed and well written study. The number of eyes considered for the study, although apparently small, allowed nonetheless to reach statistically significant results. 

Author Response

The study of Costanzo et al shows that, by taking into account a number of OCT and OCTA parameters and after setting up specific binary logistic regression to analyze OCT, OCTA and OCT/OCTA-based models, it was possible to identify a pool of baseline biomarkers of use for the prediction of responsiveness to dexamethasone implant treatment.

Comments to the manuscript

This a well designed, well performed and well written study. The number of eyes considered for the study, although apparently small, allowed nonetheless to reach statistically significant results. 

R: We would like to thank the reviewer for appreciating our work, hoping that these data could be useful in future studies with a larger number of patients or to detect specific models for artificial intelligence software.